# Design and Synthesis of Novel Chalcone Derivatives: Anti-Breast Cancer Activity Evaluation and Docking Study

**DOI:** 10.3390/ijms242115549

**Published:** 2023-10-25

**Authors:** Weihong Lai, Jiaxin Chen, Xinjiao Gao, Xiaobao Jin, Gong Chen, Lianbao Ye

**Affiliations:** 1School of Pharmacy, Guangdong Pharmaceutical University, Guangzhou 510006, China; laiweih13@163.com (W.L.); 13570784189@163.com (J.C.); crystal06252018@163.com (X.G.); 2Guangdong Key Laboratory of Pharmaceutical Bioactive Substances, Guangdong Pharmaceutical University, Guangzhou 510006, China; jinxf2001@163.com

**Keywords:** chalcones, synthesis, anti-breast cancer (MCF-7), molecular docking

## Abstract

Chalcone is a common simple fragment of natural products with anticancer activity. In a previous study, the research group discovered a series of chalcone derivatives with stronger anticancer activities. To find better anticancer drugs, novel chalcone derivatives **A1**–**A14**, **B1**–**B14** have continuously been designed and synthesized. The antiproliferative activity of these compounds against breast cancer cells (MCF-7) was investigated by the Cell Counting Kit-8 (CCK-8) method with 5-fluorouracil (5-Fu) as the control drug. The results showed that compound **A14** exhibited excellent antiproliferative ability compared to the control drug 5-Fu. Scratch experiments and cloning experiments further confirmed that compound **A14** could inhibit the proliferation and colony formation activity of MCF-7 cells. In addition, molecular docking primarily explains the interaction between compound and protein. These results suggested that compound **A14** could be a promising chalcone derivative for further anti-breast cancer research.

## 1. Introduction

Breast cancer is still one of the most common cancers among women worldwide [1,2]; though the cure rate of breast cancer has improved in the past decades due to early diagnosis [3] and surgery [4,5], the effectiveness of chemotherapy has not improved significantly [6]. In addition, the side effects caused by existing chemotherapy drugs are still a long-term challenge that threat patient’s health [7,8]. Therefore, it is still urgently desirable to develop novel chemotherapeutic agents for breast cancer without hurting normal cells and damaging human health [9].

Natural products as lead compounds play an extremely important role in the development of new drugs [10,11,12]. Chalcone is a simple chemical scaffold commonly found in many natural products with the structure of 1,3-diphenyl-2-propene-1-one [13]. Because of the simplicity of synthesis, many chalcone derivatives with a broad range of biological activities have been developed [14,15]. In most research, a chalcone scaffold with two aromatic systems (Ring A and Ring B in Figure 1A) was synthesized by Claisen–Schmidt condensation [16]. The design methods of most chalcone derivatives mainly include (1) substitution of aromatic rings, (2) Binding with other pharmacological heterocycles, (3) Residue modification of two aryl groups [17]. Several chalcone-based compounds have been approved for marketing and clinical research [18]. For instance, Metochalcone is now used as a cholagogue [19]; Sofalcone was developed as an antiulcer and mucoprotective drug [20,21] (Figure 1A). These researches exemplify the chalcone-containing chemicals as potential drug candidate deserves more attention.

Microtubules are polymers formed by the combination of α-tubulin and β-tubulin [22,23]. Due to the kinetic characteristics of its polymerization and depolymerization, microtubules play an essential role in participating in cellular functions such as cell signaling [24], cell mitosis [25], cellular material transport [26], organelle localization and cell division [27]. Thus, microtubulin is considered an important target for cancer chemotherapy because of its ability to arrest tumor cell’s continuous abnormal proliferation [28]. It’s of significance to the development of microtubulin polymerization inhibitors for tumor chemotherapy [29,30]. Several microtubulin polymerization inhibition agents have been developed, such as Colchicine [31,32], Which is a representative microtubulin inhibitor that can lead to microtubule depolymerization [33]. In addition, Combretastatin A-4 (CA-4) acts as a microtubule destabilizing agent by specifically binding to the β-subunit of tubulin [34,35]. However, high toxicity, poor solubility, and low oral bioavailability limit its further application [36]. According to the structure features of these agents, the trimethoxy phenyl is an indispensable group for binding to the β-tubulin [37] (Figure 1B).

Based on the above-mentioned background, according to the principle of pharmacophore collocationin, we firstly retained the basic structure of chalcone as scaffold B and substituted the A-ring of chalcone structure with 3,4,5-trimethoxyphenyl to form the scaffold A. Then, we proceeded with the design and structural modification and obtained A and B-series derivatives (Figure 2). The anti-breast cancer (MCF-7) cellular activity of these compounds was evaluated by the CCK-8 method. Moreover, molecular docking was analyzed to verify whether compound **A14** inhibited tubulin polymerization by binding to the colchicine binding site.

## 2. Results and Discussion

### 2.1. Chemistry

The synthetic procedures of compounds **A1**–**A14** and **B1**–**B14** were illustrated in Figure 1 and Figure 2. Initially, acetyl groups were introduced to the starting compound 1,2,3-trimethoxybenzene (**1**) via the Friedel-Crafts reaction reaction and the 1-(3,4,5-trimethoxyphenyl) ethan-1-one (**2**) was obtained. Then, 4-formylbenzoic acid (**3**) was added into a solution of compound **2** in MeOH in the presence of an aqueous solution of sodium hydroxide to give (*E*)-4-(3-oxo-3-(3,4,5-trimethoxyphenyl)prop-1-en-1-yl)benzoic acid (**4a**), which formed the crude intermediate **5a** under oxalylchloride treatment in the presence of catalytic amounts of DMF. Finally, Terminal products **A1**–**A10** were obtained by the amide condensation of compound **5a** with the corresponding primary amine fragments in the mixture of DCM and TEA. On the other hand, intermediate **4a** was *O*-alkoxylate with alkyl bromides in the presence of K_2_CO_3_ in DMF to obtain compounds **5b**–**c**. After that, the target compounds **A11**–**A14** were obtained by nucleophilic substitution with thiol compounds. The synthesis steps of compounds **B1**–**B14** were similar to that described above. All obtained chalcone derivatives were characterized by ^1^H NMR, ^13^C NMR, and mass spectrum (ESI).

### 2.2. Biological Evaluation

#### 2.2.1. In Vitro Antiproliferative Activity

Most of the synthesized chalcone derivatives showed a stronger anti-cell proliferation ability against breast cancer tumor cells than the control drug 5-Fu (IC_50_ = 168.6 μM) (Table 1). The activity of series **A** derivatives with 3,4,5-trimethoxyphenyl group was mostly better than that of series **B**, indicating that 3,4,5-trimethoxyphenyl is an important pharmacophore fragment of chalcone derivatives to maintain the activity.

Various amide compounds **A1**–**A10** and **B1**–**B11** were introduced to aromatic rings or alicyclic in order to investigate whether the change of substituent at the C4 position on the phenyl group affects the activity. The IC_50_ values of **A1**–**A5** and **B1**–**B6** showed that the introduction of a benzene ring performed limited influence on the MCF-7 cells inhibitory activity. It was speculated that the rigid structure of the benzene ring may lead to enhanced hydrophobicity and thus reduced their potency.

However, the activity of compounds **A7**–**A9** and **B8**–**B10** with IC_50_ values range from 10.63 μM to 35.95 μM, which displayed more potency than 5-Fu, suggesting that the introduction of different heterocycles at the C4 position on the phenyl group could enhance the activity. Among them, compounds **A7** (IC_50_ = 19.07 μM), **A8** (IC_50_ = 16.72 μM), and **A9** (IC_50_ = 19.63 μM) showed more prominent tumor activity compared to that of the other compounds. 

Moreover, further conformational investigations were carried out by varying the length of oxygen alkylation and changing to the sulfhydryl substituent groups. Results showed that both modified compounds effectively inhibited MCF-7 cells, indicating that oxygen alkylation at the C4 position on the phenyl group could enhance the activity to some extent. Among these compounds, compound **A14** (IC_50_ = 9.795 μM) showed a remarkable antiproliferative effect on MCF-7 cells, whose IC_50_ value was superior to that of the positive control 5-Fu.

#### 2.2.2. Cytotoxicity of Compounds **A7**, **A8**, **A9,** and **A14** against Normal Human Liver Cell

We have selected compounds **A7**, **A8**, **A9,** and **A14** with relatively low IC_50_ values, which are considered to have more potential and are more meaningful for further research. The toxic effects of compounds **A7, A8, A9,** and **A14** on normal hepatocytes cells L-O2 were assessed in this work to examine the selective toxicity of the new compounds on cancer cells without harming normal cells. The selectivity index was calculated as a ratio of L-02 IC50/MCF-7 IC50 for these four compounds. According to the IC_50_ values and selectivity index in Table 2, all four tested compounds have significantly lower IC**_50_** against MCF-7 cells compared to those against normal hepatocyte cells. Compound A14 was the most selective MCF-7 cell inhabitor; compound A8 showed the next highest level of potency and selectivity for MCF-7 cells. Therefore, **A8** and **A14** were chosen as representative medications for future research after thorough evaluation.

#### 2.2.3. Migration Ability Assay

An obvious feature of breast cancer cells is their strong migration ability. In order to study the inhibitory effect of compounds on breast cancer cells migration, the scratch distances of compounds **A8** and **A14** on MCF-7 cells at different concentrations were measured by wound healing experiments. Figure 3 and Figure 4 showed that compared with the control group, compounds **A8** and **A14** effectively inhibited the migration of MCF-7 cells in a dose-dependent manner. Moreover, under similar concentrations, the inhibitory migration effect of the target compound on MCF-7 is more excellent than that of the control drug 5-Fu.

#### 2.2.4. Cell Colony Formatting Assay

To further assess the effect of the most active optimal compound, **A14**, on the proliferation of MCF-7 cells, a cell colony formation assay was performed after 7 days of treatment. Colonies with drug concentrations greater than 8 µmol/L were significantly reduced compared to the blank control (Figure 5). The results indicated that compound **A14** possesses colony-formation inhibitory abilities against MCF-7 cells. 

### 2.3. Molecular Docking Study

Previous studies indicated that 3,4,5-trimethoxyphenyl unit as a potent fragment to interact with tubulin [38]. To better understand the inhibitory mechanism of representative compound **A14** on breast cancer cells, docking studies of compound **A14** were performed at the colchicine binding site of the tubulin crystal structure (PDB ID: 4O2B) utilizing the AutoDock software (http://vina.scripps.edu/, accessed on 18 March 2023). The binding free energy of compound **A14** was −7.2 kcal/mol, and the binding free energy of CA-4 was −6.0 kcal/mol, indicating the binding affinity of compound A14 towards tubulin. As shown in Figure 6, docking studies reveal that salt bridges and hydrogen bonding interactions play a key role in binding. The oxygen atom in 3,4,5-trimethoxyphenyl of compound **A14** forms a hydrogen bonding interaction with hydrogen in secondary amine of Arg-61 (2.3 Å), the carbonyl of α,β-unsaturated carbonyl system establishes hydrogen bonds with hydrogen atom in primary amine of Arg 164 (2.7 Å) and Arg 105 (2.1 Å). In general, these findings suggested that compound **A14** bearing 3,4,5-trimethoxyphenyl exhibited favorable docking with tubulin, implying that compounds’ potential to hinder the proliferation of breast cancer cells may be related to their inhibition of tubulin.

## 3. Materials and Methods

### 3.1. Materials

#### 3.1.1. Chemistry

All chemicals were purchased from Macklin or Aladdin and were dried and purified by traditional methods before use. Silica gel 60F-254 (Merck KGaA, Darmstadt, Germany). Silica plates were ordered for Thin-Layer Chromatography (TLC) plates, and silica gel 60–120 mesh was used for column chromatography. Nuclear magnetic resonance (NMR) spectra were recorded on a superconducting NMR instrument (Bruker AVANCE AV 400, Karlsruhe, Germany) with TMS as the internal standard, DMSO-*d_6_* (2.5 ppm ^1^H, 39.5 ppm ^13^C) and CDCl_3_ (7.26 ppm ^1^H, 77 ppm ^13^C) as solvent. The chemical structures of the target compounds were determined by Electron Impact Mass Spectra (EI-MS) using a mass spectrometer (Agilent, Santa Clara, CA, USA). Melting points (m.p.) of all final compounds were recorded on an SGW X-4 apparatus (Shanghai, China). Synthesized compounds were analyzed by the high-performance liquid chromatography (HPLC) instrument (Agilent, Santa Clara, CA, USA), using the following column: 5 μm, 4.6 mm × 250 mm Supersil ODS2 C18 column.

#### 3.1.2. Biological Evaluation

DMEM high glucose medium and fetal calf serum were obtained from Gibco (New York, NY, USA), and cell counting Kit-8 solution was purchased from Meilunbio (Dalian, China). Paraformaldehyde and Crystal Violet were purchased from Biosharp (Hefei, China). The MCF-7 and L-02 cell lines were purchased from the Shanghai Institute of Biochemistry (Shanghai, China) All the target compounds and 5-fluorouracil (5-Fu) were dissolved with DMSO at a concentration of 20 mM. Cell culture medium was a mixture of DMEM medium, 10% fetal bovine serum, and 1% penicillin–streptomycin solution stored at −4 °C. The cells were grown at 37 °C under 5% CO_2_.

### 3.2. Synthesis of Compounds

*Synthesis of 1-(3,4,5-trimethoxyphenyl)ethanone* (**2a**)

To a solution of 1,2,3-trimethoxybenzene(**1**) (1.7 g, 10 mmol) in anhydrous DCM (40 mL), was added anhydrous aluminum trichloride (2.6 g, 20 mmol) and stirred together. Then, anhydrous acetyl chloride (0.73 mL, 10 mmol) was slowly added to the mixture. The reaction was protected by nitrogen gas and stirred at 0 °C for 4 h. Reaction mixture was quenched with water, extracted with DCM, and dried with Na_2_SO_4_. The organic phase was concentrated under reduced pressure. The crude product was purified to obtain the intermediate **2a** through silica gel column chromatography.

*Synthesis of (E)-4-(3-oxo-3-(3,4,5-trimethoxyphenyl)prop-1-en-1-yl)benzoic acid* (**4a**)

To a solution of compound **2a** (2.1 g, 10 mmol) and 4-formylbenzoic acid (1.52 g, 10.1 mmol) in methanol (35 mL), 50% sodium hydroxide solution (4 mL) was slowly added, and the reaction solution was stirred at 50 °C for 5 h under reflux condenser. After completion of the reaction, the mixture was transferred to room temperature for cooling. Subsequently, an appropriate amount of dilute hydrochloric acid (0.1 mol/L) was added until the pH of the reaction solution was 3–4, and a large amount of solid was precipitated. The mixture was then filtered, and the filter cake was washed with methanol multiple times and dried to afford product **4a**.

*Preparations of compound* **4b** *were similar to* **4a**.*Synthesis of (E)-4-(3-oxo-3-(3,4,5-trimethoxyphenyl)prop-1-en-1-yl) benzoyl chloride* (**5a**)

Compound **4a** (1.7 g, 5 mmol) was dissolved in anhydrous DCM (20 mL) and cooled to 0 °C. Next, anhydrous oxalyl chloride (0.6 mL, 6 mmol) was slowly added dropwise, followed by 2 drops of *N*, *N*-dimethylformamide as a catalyst. The reaction flask was then transferred to room temperature and left to react for 12 h. The progress of the reaction was monitored with TLC(DCM/MeOH 50:1). Crude product of yellow solid **5a** was obtained via depressurization and concentration.

*Preparations of compound* **5d** *were similar to* **5a**.*Synthesis of 2-bromoethyl (E)-4-(3-oxo-3-(3,4,5-trimethoxyphenyl)Prop-1-en-1-yl)benzoate* (**5b**)

To a solution of compound **4a** (1.7 g, 5 mmol) in *N*,*N*-dimethylformamide (10 mL) at room temperature, dibromoethane (0.7 mL, 8 mmol, ρ = 2.18 g/cm^3^) was slowly added and reacted for 8 h. After completion of the reaction, the mixture was extracted with ethyl acetate 3–4 times; the organic layer was collected and dried with anhydrous sodium sulfate. The organic solution was concentrated under reduced pressure to obtain a crude product. Subsequently, compound **5b** was purified by silica gel column chromatography.

*Preparations of compounds* **5c**, **5e**, *and* **5f** *were similar to* **5b**.
*Synthesis of (E)-N-(4-fluorophenyl)-4-(3-oxo-3-(3,4,5-trimethoxyphenyl)prop-1-en-1-yl)*
*benzamide* (**A1**)

To a solution of crude product **5a** in dichloromethane was added anhydrous triethylamine (0.7 mL, 5 mmol) and 4-fluoroaniline (0.55 g, 5 mmol), it was reacted overnight under anhydrous conditions at room temperature. After the reaction had finished, the mixture was washed with water and extracted with dichloromethane 3–4 times, and the lower organic layer was collected and dried with anhydrous sodium sulfate. The solution was concentrated under reduced pressure to obtain a crude product. Compound **A1** was further purified through silica gel column chromatography. 

*Preparations of compounds* **A2**–**A10**, **B1**–**B11** *were similar to* **A1**.*Synthesis of 2-((1H-benzo[d]imidazol-2-yl)thio)ethyl (E)-4-(3-oxo-3-(3,4,5-trimethoxyphenyl)prop-1-en-1-yl)benzoate* (**A11**)

To a solution of 1H-benzo[d]imidazole-2-thiol (0.122 g, 1.1 mmol) in *N*, *N*-dimethylformamide (10 mL) was added anhydrous potassium carbonate (0.13 g, 1.1 mmol) and compound **5b** (0.45 g, 1 mmol). The reaction mixture was stirred at room temperature for 8 h. Upon completion of the reaction, the mixture was washed with saturated salt water and extracted with ethyl acetate 3–4 times. The organic layer was collected and dried with anhydrous sodium sulfate. The solution was concentrated under reduced pressure to obtain a crude product. Next, the final product **A11** was purified through silica gel column chromatography.

*Preparations of compounds* **A11**–**A14**, **B12**–**B14** *was similar to* **A11**.

(*E*)-4-(3-oxo-3-(3,4,5-trimethoxyphenyl) prop-1-en-1-yl) benzoic acid (**2a**). Eluent petroleum ether/EtOAc (10:1). Yellow solid, 78% yield. mp: 165.5–168.3 °C. ^1^H NMR (400 MHz, DMSO-*d_6_*) δ 8.10–7.96 (m, 5H), 7.79 (d, *J* = 15.6 Hz, 1H), 7.46 (s, 2H), 3.92 (s, 6H), 3.78 (s, 3H).^13^C NMR (101 MHz, DMSO-*d_6_*) δ 198.47, 153.39, 143.95, 131.28, 107.96, 61.07, 56.81, 28.21 (Appendix A).

(*E*)-4-(3-oxo-3-phenylprop-1-en-1-yl) benzoic acid (**4a**). Yellow solid, 80% yield. mp: 187.3–190.5 °C. ^1^H NMR (400 MHz, DMSO-*d_6_*) δ 8.18 (d, *J* = 7.1 Hz, 1H), 8.08–8.00 (m, 2H), 7.79 (d, *J* = 15.7 Hz, 1H), 7.69 (t, *J* = 7.4 Hz, 1H), 7.59 (t, *J* = 7.6 Hz, 1H). ^13^C NMR (101 MHz, DMSO-*d_6_*) δ 191.17, 172.49, 153.13, 145.52, 144.05, 138.70, 132.03, 130.88, 130.82, 130.66, 127.15, 107.63, 61.07, 56.81. ESI-MS *m/z*: 342.86 [M + H]^+^, C_19_H_18_O_6_ [342.11] (Appendix A). 

(*E*)-4-(3-oxo-3-phenylprop-1-en-1-yl) benzoic acid (**4b**). Yellow solid, 80% yield. mp: 178.5–181.6 °C. ^1^H NMR (400 MHz, DMSO-*d_6_*) δ 8.18 (d, *J* = 7.1 Hz, 1H), 8.08–8.00 (m, 2H), 7.79 (d, *J* = 15.7 Hz, 1H), 7.69 (t, *J* = 7.4 Hz, 1H), 7.59 (t, *J* = 7.6 Hz, 1H).^13^C NMR (101 MHz, DMSO-*d_6_*) δ 192.81, 172.49, 145.52, 139.53, 138.02, 133.52, 132.03, 131.68, 131.42, 130.88, 130.82, 126.34. ESI-MS *m/z*: 253.26 [M + H]^+^, C_16_H_12_O_3_ [252.08] (Appendix A).

2-bromoethyl (*E*)-4-(3-(3,5-dimethoxy-4-(l1-oxidaneyl) phenyl)-3-oxoprop-1-en-1-yl) benzoate (**5b**). Eluent petroleum ether/EtOAc (2:1). Yellow solid, 70% yield. mp: 189.1–192.3 °C. ^1^H NMR (400 MHz, DMSO-*d_6_*) δ 8.10–8.02 (m, 5H), 7.78 (d, *J* = 15.6 Hz, 1H), 7.45 (s, 2H), 4.65–4.59 (m, 2H), 3.91 (s, 6H), 3.87–3.83 (m, 2H), 3.78 (s, 3H). ^13^C NMR (101 MHz, DMSO-*d_6_*) δ 191.17, 169.53, 153.13, 145.52, 144.05, 138.70, 133.32, 131.20, 130.79, 130.66, 127.15, 107.63, 67.21, 61.07, 56.81, 32.70, 31.38.ESI-MS *m/z*: 449.22 [M + H]^+^, C_21_H_21_B_r_O_6_ [448.05] (Appendix A).

3-bromopropyl(*E*)-4-(3-(3,5-dimethoxy-4-(l1-oxidaneyl) phenyl)-3-oxoprop-1-en-1-yl) benzoate (**5c**). Eluent petroleum ether/EtOAc (1:1). Yellow solid, 71% yield. mp: 160.5–162.9 °C. ^1^H NMR (400 MHz, DMSO-*d_6_*) δ 8.10–8.02 (m, 5H), 7.79 (d, *J* = 15.6 Hz, 1H), 7.45 (s, 2H), 4.37 (dt, *J* = 21.8, 6.1 Hz, 2H), 3.91 (s, 6H), 3.78 (s, 3H), 3.70 (dd, *J* = 13.7, 7.2 Hz, 2H), 2.27 (dp, *J* = 12.3, 6.3 Hz, 2H). ^13^C NMR (101 MHz, DMSO-*d_6_*) δ 191.17, 169.53, 153.13, 145.52, 144.05, 138.70, 133.32, 131.20, 130.79, 130.66, 127.15, 107.63, 67.21, 61.07, 56.81, 32.70, 31.38. ESI-MS *m/z*: 463.19 [M + H]^+^, C_22_H_23_B_r_O_6_ [462.07] (Appendix A).

2-bromoethyl (*E*)-4-(3-oxo-3-phenylprop-1-en-1-yl) benzoate (**5e**). Eluent petroleum ether/EtOAc (2:1). Yellow solid, 65% yield. mp: 193.4–196.9 °C. ^1^H NMR (400 MHz, DMSO-*d_6_*) δ 8.18 (d, *J* = 7.4 Hz, 2H), 8.11–8.01 (m, 6H), 7.79 (d, *J* = 15.7 Hz, 1H), 7.70 (t, *J* = 7.3 Hz, 1H), 7.60 (t, *J* = 7.6 Hz, 2H), 4.63 (t, *J* = 5.5 Hz, 2H), 3.85 (t, *J* = 5.5 Hz, 2H). ^13^C NMR (101 MHz, DMSO-*d*) δ 192.81, 169.71, 145.52, 139.53, 138.02, 133.52, 132.65, 131.68, 131.42, 131.20, 130.91, 126.34, 66.77, 30.61. ESI-MS *m/z*: 359.11 [M + H]^+^, C_18_H_15_B_r_O_3_ [358.02] (Appendix A).

3-bromopropyl (*E*)-4-(3-oxo-3-phenylprop-1-en-1-yl) benzoate (**5f**). Eluent petroleum ether/EtOAc (2:1). Yellow solid, 62% yield. mp: 200.4–204.7 °C. ^1^H NMR (400 MHz, DMSO-*d_6_*) δ 8.20–8.16 (m, 2H), 8.04 (s, 5H), 7.79 (d, *J* = 15.7 Hz, 1H), 7.70 (t, *J* = 7.3 Hz, 1H), 7.60 (t, *J* = 7.7 Hz, 2H), 4.40 (t, *J* = 6.1 Hz, 2H), 3.69 (t, *J* = 6.5 Hz, 2H), 2.29 (p, *J* = 6.3 Hz, 2H). ^13^C NMR (125 MHz, DMSO-*d_6_*) δ 192.81, 169.53, 145.52, 139.53, 138.02, 133.52, 133.32, 131.68, 131.42, 131.20, 130.79, 126.34, 67.21, 32.70, 31.38. ESI-MS *m/z*: 373.07 [M + H]^+^, C_19_H_17_B_r_O_3_ [372.04] (Appendix A).

(*E*)*-N-*(4-fluorophenyl)-4-(3-oxo-3-(3,4,5-trimethoxyphenyl) prop-1-en-1-yl) benzamide (**A1**). Eluent petroleum ether/EtOAc (6:1). Yellow solid, 40% yield, mp: 230.6–232.8 °C. ^1^H NMR (400 MHz, DMSO-*d_6_*) δ 8.11–8.03 (m, 2H), 7.89–7.71 (m, 5H), 7.50 (d, *J* = 8.4 Hz, 1H), 7.47 (s, 1H), 7.28–7.15 (m, 3H), 3.92 (s, 3H), 3.86–3.77 (m, 6H). ^13^C NMR (101 MHz, DMSO-*d_6_*) δ 197.73, 188.33, 165.26, 153.44, 152.97, 143.10, 138.16, 136.51, 133.27, 129.37, 128.59, 127.72, 124.07, 122.72, 115.81, 115.71, 115.59, 115.49, 106.79, 106.22, 60.69, 60.59, 56.74, 56.53, 55.37. ESI-MS *m/z*: 435.92 [M + H]^+^, C_25_H_22_FNO_5_ [435.15] (Appendix A).

(*E*)*-N-*(4-bromophenyl)-4-(3-oxo-3-(3,4,5-trimethoxyphenyl) prop-1-en-1-yl) benzamide (**A2**). Eluent petroleum ether/EtOAc (7:1). Yellow solid, 51% yield, mp: 158.4–160.8 °C. ^1^H NMR (400 MHz, DMSO-*d_6_*) δ 8.13–8.00 (m, 3H), 7.89–7.82 (m, 1H), 7.76 (dd, *J* = 19.4, 8.8 Hz, 2H), 7.56 (td, *J* = 16.6, 15.4, 8.5 Hz, 3H), 7.46 (s, 1H), 7.26 (d, *J* = 7.4 Hz, 1H), 7.13 (d, *J* = 8.7 Hz, 1H), 6.50 (t, *J* = 8.3 Hz, 1H), 3.88 (d, *J* = 30.7 Hz, 6H), 3.76 (d, *J* = 18.2 Hz, 3H). ^13^C NMR (101 MHz, DMSO-*d_6_*) δ 191.17, 169.54, 153.13, 145.52, 144.05, 141.21, 137.87, 137.56, 132.20, 131.67, 131.08, 130.66, 127.15, 123.40, 107.63, 90.49, 61.07, 56.81. ESI-MS *m/z*: 517.75 [M + Na]^+^, C_25_H_22_BrNO_5_ [495.07] (Appendix A).

(*E*)*-N-*(4-iodophenyl)-4-(3-oxo-3-(3,4,5-trimethoxyphenyl) prop-1-en-1-yl) benzamide (**A3**). Eluent petroleum ether/EtOAc (6:1). Brown solid, 45% yield, mp: 210.5–212.3 °C. ^1^H NMR (400 MHz, DMSO-*d_6_*) δ 10.34 (d, *J* = 68.4 Hz, 2H), 8.10–8.05 (m, 1H), 8.03 (d, *J* = 8.3 Hz, 1H), 7.85 (d, *J* = 8.2 Hz, 1H), 7.69 (dd, *J* = 15.7, 8.8 Hz, 3H), 7.59 (dd, *J* = 8.6, 3.2 Hz, 3H), 7.29–7.21 (m, 3H), 3.92 (s, 3H), 3.81 (d, *J* = 24.3 Hz, 6H). ^13^C NMR (101 MHz, DMSO-*d_6_*) δ 191.17, 169.54, 153.13, 145.52, 144.05, 141.21, 137.87, 137.56, 132.20, 131.67, 131.08, 130.66, 127.15, 123.40, 107.63, 90.49, 61.07, 56.81. ESI-MS *m/z*: 565.82 [M + Na]^+^, C_25_H_22_INO_5_ [543.05]. (Appendix A).

(*E*)*-N-*(4-methoxyphenyl)-4-(3-oxo-3-(3,4,5-trimethoxyphenyl) prop-1-en-1-yl) benzamide (**A4**). Eluent petroleum ether/EtOAc (6:1). Brown solid, 39% yield, mp: 205.3–207.6 °C. ^1^H NMR (400 MHz, Chloroform-d) δ 7.96 (s, 1H), 7.91 (d, *J* = 7.5 Hz, 2H), 7.80 (d, *J* = 15.5 Hz, 1H), 7.71 (d, *J* = 7.6 Hz, 2H), 7.57 (d, *J* = 8.4 Hz, 2H), 7.28 (s, 2H), 6.91 (d, *J* = 7.9 Hz, 2H), 3.95 (s, 9H), 3.81 (s, 3H). ^13^C NMR (101 MHz, Chloroform-d) δ 191.17, 169.54, 159.47, 153.13, 145.52, 144.05, 137.56, 136.26, 132.20, 131.67, 131.08, 130.66, 127.15, 121.98, 116.72, 107.63, 61.07, 56.81, 55.86. ESI-MS *m/z*: 447.93 [M + H]^+^, C_26_H_25_NO_6_ [447.17] (Appendix A).

(*E*)-4-(3-oxo-3-(3,4,5-trimethoxyphenyl) prop-1-en-1-yl)*-N-*(p-tolyl) benzamide (**A5**)**.** Eluent petroleum ether/EtOAc (5:1). Yellow solid, 49% yield, mp: 239.7–241.4 °C. ^1^H NMR (400 MHz, DMSO-*d_6_*) δ 10.29–10.05 (m, 1H), 8.11–8.01 (m, 3H), 7.97–7.76 (m, 2H), 7.70–7.55 (m, 3H), 7.47 (s, 1H), 7.26 (d, *J* = 9.4 Hz, 1H), 7.15 (dd, *J* = 17.2, 6.2 Hz, 2H), 3.92 (s, 3H), 3.86–3.72 (m, 6H), 2.28 (d, J = 7.1 Hz, 3H).^13^C NMR (101 MHz, DMSO-*d_6_*) δ 191.17, 169.54, 153.13, 145.52, 144.05, 140.24, 137.56, 135.70, 132.20, 131.70, 131.67, 131.08, 130.66, 127.15, 119.62, 107.63, 61.07, 56.81, 21.89. ESI-MS *m/z*: 431.96 [M + H]^+^, C_26_H_25_NO_5_ [431.17] (Appendix A).

(*E*)-4-(3-oxo-3-(3,4,5-trimethoxyphenyl) prop-1-en-1-yl)*-N-*(thiazol-2-yl) benzamide (**A6**). dichloromethane/methanol(70:1). Yellow solid, 70% yield, mp: 260.6–263.9 °C. ^1^H NMR (400 MHz, Chloroform-d) δ 8.09 (d, *J* = 7.9 Hz, 2H), 7.79 (d, *J* = 7.8 Hz, 2H), 7.60 (d, *J* = 15.7 Hz, 1H), 7.31 (s, 2H), 7.27 (s, 1H), 7.08 (s, 1H), 7.00 (d, *J* = 3.0 Hz, 1H), 3.96 (d, *J* = 4.2 Hz, 9H).^13^C NMR (101 MHz, Chloroform-d) δ 191.17, 168.62, 168.57, 153.13, 145.52, 144.05, 139.07, 137.56, 131.80, 131.08, 130.66, 130.66, 127.15, 113.96, 107.63, 61.07, 56.81. ESI-MS *m/z*: 424.81 [M + H]^+^, C_22_H_20_N_2_O_5_S [424.11] (Appendix A).

(*E*)-4-(3-oxo-3-(3,4,5-trimethoxyphenyl) prop-1-en-1-yl)-*N*-(thiophen-2-ylmethyl) benzamide (**A7**). dichloromethane/methanol(80:1). Yellow solid, 60% yield, mp: 156.7–159.2 °C. ^1^H NMR (400 MHz, DMSO-*d_6_*) δ 9.26 (t, *J* = 5.9 Hz, 1H), 8.09–8.00 (m, 3H), 7.96 (d, *J* = 8.3 Hz, 2H), 7.78 (d, *J* = 15.5 Hz, 1H), 7.46 (s, 2H), 7.40 (dd, *J* = 5.0, 1.1 Hz, 1H), 7.04 (d, *J* = 2.4 Hz, 1H), 6.98 (dd, *J* = 4.9, 3.6 Hz, 1H), 4.66 (d, *J* = 5.8 Hz, 2H), 3.92 (s, 6H), 3.78 (s, 3H). ^13^C NMR (101 MHz, DMSO-*d_6_*) δ 188.29, 165.83, 153.43, 143.15, 143.01, 142.62, 137.92, 135.84, 133.28, 129.38, 128.19, 127.12, 125.94, 125.50, 123.88, 106.77, 60.68, 56.73, 55.38. ESI-MS *m/z*: 437.91 [M + H]^+^, C_24_H_23_NO_5_S [437.13] (Appendix A).

(*E*)-4-(3-oxo-3-(3,4,5-trimethoxyphenyl) prop-1-en-1-yl)*-N-*((tetrahydrofuran-3-yl) methyl) benzamide (**A8**). dichloromethane/methanol(100:1). Yellow solid, 41% yield, mp: 141.5–142.9 °C. ^1^H NMR (400 MHz, DMSO-*d_6_*) δ 8.62 (dt, *J* = 50.7, 5.6 Hz, 1H), 8.04–7.99 (m, 1H), 7.93 (d, *J* = 8.3 Hz, 1H), 7.75 (d, *J* = 8.4 Hz, 1H), 7.46 (d, *J* = 5.0 Hz, 2H), 7.25 (s, 1H), 7.14–6.91 (m, 1H), 3.92 (s, 3H), 3.82–3.73 (m, 6H), 3.65 (ddd, *J* = 26.7, 11.6, 5.8 Hz, 2H), 3.48 (ddd, *J* = 16.3, 8.4, 5.3 Hz, 1H), 3.32–3.19 (m, 2H), 1.97–1.90 (m, 1H), 1.61 (tt, *J* = 13.3, 6.8 Hz, 1H), 1.26 (t, *J* = 16.0 Hz, 1H). ^13^C NMR (101 MHz, DMSO-*d_6_*) δ 192.95, 188.30, 166.26, 166.21, 153.42, 153.31, 143.21, 142.71, 142.61, 138.42, 138.26, 137.70, 136.32, 134.75, 133.29, 132.28, 129.37, 129.28, 128.95, 128.12, 127.57, 123.76, 106.76, 106.62, 70.97, 70.95, 67.27, 67.25, 60.66, 60.62, 56.71, 56.45, 55.35, 42.46, 42.39, 29.98, 29.95. ESI-MS *m/z*: 425.97 [M + H]^+^, C_24_H_27_NO_6_ [425.18] (Appendix A).

(*E*)*-N-*(cyclopropylmethyl)-4-(3-oxo-3-(3,4,5-trimethoxyphenyl) prop-1-en-1-yl) benzamide (**A9**). dichloromethane/methanol(100:1). White solid, 69% yield, mp: 155.6–156.3 °C. ^1^H NMR (400 MHz, DMSO-*d_6_*) δ 8.43 (t, *J* = 5.6 Hz, 1H), 7.83–7.74 (m, 3H), 7.70 (d, *J* = 8.3 Hz, 2H), 7.54 (d, *J* = 15.6 Hz, 1H), 7.22 (s, 2H), 3.68 (s, 6H), 3.54 (s, 3H), 2.93 (t, *J* = 6.2 Hz, 2H), 0.88–0.75 (m, 1H), 0.24–0.18 (m, 2H), 0.01 (q, *J* = 4.8 Hz, 2H). ^13^C NMR (101 MHz, DMSO-*d_6_*) δ 188.30, 165.89, 153.42, 143.24, 142.60, 137.62, 136.42, 133.29, 129.28, 128.13, 123.71, 106.76, 60.67, 56.72, 55.37, 44.07, 11.48, 3.82. ESI-MS *m/z*: 395.87 [M + H]^+^, C_23_H_25_NO_5_ [395.17] (Appendix A).

(*E*)-4-(3-oxo-3-(3,4,5-trimethoxyphenyl) prop-1-en-1-yl)*-N-*(1H-pyrazol-5-yl) benzamide (**A10**). petroleum ether/EtOAc (6:1). Brown solid, 38% yield, mp: 103.7–105.7 °C. ^1^H NMR (400 MHz, DMSO-*d_6_*) δ 9.38 (s, 1H), 8.05–8.00 (m, 2H), 7.80 (t, *J* = 1.0 Hz, 1H), 7.65–7.61 (m, 3H), 7.47 (s, 1H), 7.23 (s, 2H), 6.61 (s, 1H), 3.86 (s, 6H), 3.78 (s, 3H).^13^C NMR (101 MHz, DMSO-*d_6_*) δ 191.17, 168.83, 153.95, 153.13, 145.52, 144.05, 137.56, 137.39, 131.73, 131.66, 131.08, 130.66, 127.15, 107.63, 97.71, 61.07, 56.81. ESI-MS *m/z*: 429.90 [M + Na]^+^, C_22_H_21_N_3_O_5_ [407.15] (Appendix A).

2-((1H-benzo[d]imidazol-2-yl) thio) ethyl (*E*)-4-(3-oxo-3-(3,4,5-trimethoxyphenyl) prop-1-en-1-yl) benzoate (**A11**). petroleum ether/EtOAc (4:1). Yellow solid, 69% yield, mp: 98.0–99.9 °C. ^1^H NMR (400 MHz, Chloroform-d) δ 7.97 (d, *J* = 8.3 Hz, 1H), 7.89–7.76 (m, 1H), 7.64–7.55 (m, 1H), 7.52 (s, 2H), 7.33 (d, *J* = 14.8 Hz, 2H), 7.28 (s, 1H), 7.23 (s, 1H), 7.20 (td, *J* = 6.9, 3.1 Hz, 3H), 4.65 (dt, *J* = 28.6, 6.4 Hz, 2H), 4.00–3.83 (m, 9H), 3.66 (dt, *J* = 23.4, 6.4 Hz, 2H). ^13^C NMR (101 MHz, Chloroform-d) δ 191.17, 170.30, 155.44, 153.13, 145.52, 144.05, 142.19, 140.51, 138.70, 132.65, 131.20, 130.91, 130.66, 127.78, 127.54, 127.15, 119.34, 116.12, 107.63, 65.59, 61.07, 56.81, 33.85. ESI-MS *m/z*: 518.97 [M + H]^+^, C_28_H_26_N_2_O_6_S [518.15] (Appendix A).

3-((1H-benzo[d]imidazol-2-yl) thio) propyl(*E*)-4-(3-oxo-3-(3,4,5-trimethoxyphenyl) prop-1-en-1-yl) benzoate (**A12**). petroleum ether/EtOAc (3:1). Yellow solid, 61% yield, mp: 96.4–98.4 °C. ^1^H NMR (400 MHz, Chloroform-d) δ 8.07–7.99 (m, 2H), 7.80 (d, *J* = 15.6 Hz, 1H), 7.64 (d, *J* = 8.1 Hz, 2H), 7.60–7.53 (m, 1H), 7.50 (dt, *J* = 6.0, 3.5 Hz, 2H), 7.29 (d, *J* = 7.9 Hz, 2H), 7.20–7.14 (m, 2H), 4.44 (t, *J* = 5.4 Hz, 2H), 3.95 (d, *J* = 4.0 Hz, 9H), 3.48 (t, *J* = 6.9 Hz, 2H), 2.32–2.22 (m, 2H). ^13^C NMR (101 MHz, Chloroform-d) δ 188.95, 165.89, 162.80, 153.21, 149.94, 143.17, 142.82, 139.16, 133.12, 131.37, 130.15, 128.29, 123.84, 122.24, 106.29, 63.47, 61.01, 56.45, 53.49, 36.62, 29.19, 28.77. ESI-MS *m/z*: 532.98 [M + H]^+^, C_29_H_28_N_2_O_6_S [532.17] (Appendix A).

2-(pyridin-4-ylthio) ethyl (*E*)-4-(3-oxo-3-(3,4,5-trimethoxyphenyl) prop-1-en-1-yl) benzoate (**A13**). petroleum ether/EtOAc (3:1). Yellow solid, 68% yield, mp: 99.6–101.2 °C. ^1^H NMR (400 MHz, Chloroform-d) δ 8.49–8.41 (m, 2H), 8.05 (d, *J* = 8.3 Hz, 1H), 7.95–7.83 (m, 1H), 7.72 (d, *J* = 8.3 Hz, 1H), 7.57 (d, *J* = 15.7 Hz, 1H), 7.42 (d, *J* = 8.3 Hz, 1H), 7.33–7.18 (m, 5H), 4.57 (dt, *J* = 26.0, 6.9 Hz, 2H), 4.01–3.84 (m, 9H), 3.39 (dt, *J* = 19.9, 6.9 Hz, 2H). ^13^C NMR (101 MHz, Chloroform-d) δ 191.17, 170.30, 153.13, 151.59, 145.52, 144.05, 138.70, 138.67, 132.65, 131.20, 130.91, 130.66, 127.15, 122.83, 107.63, 65.84, 61.07, 56.81, 34.78. ESI-MS *m/z*: 480.05 [M + H]^+^, C_26_H_25_NO_6_S [479.14] (Appendix A).

3-(pyridin-4-ylthio) propyl (*E*)-4-(3-oxo-3-(3,4,5-trimethoxyphenyl) prop-1-en-1-yl) benzoate (**A14**). petroleum ether/EtOAc (3:1). Yellow solid, 59% yield, mp: 128.9–130.3 °C. ^1^H NMR (400 MHz, Chloroform-d) δ 8.40 (d, *J* = 6.3 Hz, 2H), 8.10 (d, *J* = 8.1 Hz, 2H), 7.84–7.71 (m, 1H), 7.29 (d, *J* = 6.0 Hz, 2H), 7.16 (d, *J* = 6.4 Hz, 3H), 7.15 (d, *J* = 6.4 Hz, 2H), 4.50 (t, *J* = 6.1 Hz, 2H), 3.96 (d, *J* = 3.7 Hz, 9H), 3.16 (q, *J* = 7.6 Hz, 2H), 2.31–2.15 (m, 2H). ^13^C NMR (101 MHz, Chloroform-d) δ 188.75, 165.76, 153.25, 153.14, 149.05, 148.95, 142.94, 139.41, 133.10, 131.29, 130.16, 129.55, 129.13, 128.32, 124.00, 120.83, 106.32, 63.41, 61.00, 56.48, 56.26, 27.91, 27.39. ESI-MS *m/z*: 493.98 [M + H]^+^, C_27_H_27_NO_6_S [493.16] (Appendix A).

(*E*)*-N-*(4-fluorophenyl)-4-(3-oxo-3-phenylprop-1-en-1-yl) benzamide (**B1**). Eluent petroleum ether/EtOAc (7:1). White solid, 40% yield, mp: 177.6–179.9 °C. ^1^H NMR (400 MHz, DMSO-*d_6_*) δ 10.40 (s, 1H), 8.22–8.18 (m, 2H), 8.09–8.01 (m, 5H), 7.85–7.79 (m, 3H), 7.73–7.68 (m, 1H), 7.60 (t, *J* = 7.6 Hz, 2H), 7.21 (t, *J* = 8.9 Hz, 2H). ^13^C NMR (101 MHz, DMSO-*d_6_*) δ 189.63, 165.23, 143.23, 138.10, 137.87, 136.55, 135.88, 133.81, 129.40, 129.30, 129.10, 128.64, 128.09, 124.22, 122.78, 122.70, 115.80, 115.58 (Appendix A).

(*E*)*-N-*(4-chlorophenyl)-4-(3-oxo-3-phenylprop-1-en-1-yl) benzamide (**B2**)**.** Eluent petroleum ether/EtOAc (6:1). Yellow solid, 39% yield, mp: 148.8–151.6 °C. ^1^H NMR (400 MHz, DMSO-*d_6_*) δ 10.29 (s, 1H), 8.01–7.96 (m, 2H), 7.85 (d, *J* = 8.0 Hz, 2H), 7.83–7.76 (m, 2H), 7.65 (t, *J* = 7.4 Hz, 1H), 7.60 (d, *J* = 8.1 Hz, 2H), 7.53 (t, *J* = 7.6 Hz, 2H), 7.39 (d, *J* = 8.9 Hz, 2H), 7.01 (d, *J* = 8.8 Hz, 2H). ^13^C NMR (101 MHz, DMSO-*d_6_*) δ 192.81, 169.54, 145.52, 141.57, 139.53, 137.56, 133.52, 132.23, 132.20, 131.68, 131.67, 131.42, 131.08, 130.21, 126.34, 124.01. ESI-MS *m/z*: 361.83 [M + H]^+^, C_22_H_16_ClNO_2_ [361.09] (Appendix A).

(*E*)*-N-*(4-bromophenyl)-4-(3-oxo-3-phenylprop-1-en-1-yl) benzamide (**B3**). Eluent petroleum ether/EtOAc (6:1). White solid, 42% yield, mp: 204.5–206.2 °C. ^1^H NMR (400 MHz, DMSO-*d_6_*) δ 10.46 (s, 1H), 8.22–8.17 (m, 2H), 8.09–8.01 (m, 5H), 7.81–7.77 (m, 2H), 7.70 (t, *J* = 7.3 Hz, 1H), 7.65–7.51 (m, 5H). ^13^C NMR (101 MHz, DMSO-*d_6_*) δ 189.62, 165.42, 143.19, 138.93, 138.21, 137.86, 136.42, 133.82, 131.94, 129.31, 129.11, 128.70, 124.28, 122.76, 115.97. ESI-MS *m/z*: 405.82 [M + H]^+^, C_22_H_16_BrNO_2_ [405.04] (Appendix A).

(*E*)*-N-*(4-iodophenyl)-4-(3-oxo-3-phenylprop-1-en-1-yl) benzamide (**B4**). Eluent petroleum ether/EtOAc (5:1). Brown solid, 43% yield, mp: 198.0–200.1 °C. ^1^H NMR (400 MHz, DMSO-*d_6_*) δ 10.43 (s, 1H), 8.19 (d, *J* = 7.6 Hz, 2H), 8.05 (q, *J* = 8.3 Hz, 5H), 7.81 (d, *J* = 15.6 Hz, 1H), 7.69 (dd, *J* = 15.2, 7.9 Hz, 5H), 7.60 (t, *J* = 7.6 Hz, 2H). ^13^C NMR (101 MHz, DMSO-*d_6_*) δ 189.63, 165.41, 143.20, 139.40, 138.20, 137.86, 137.78, 136.45, 133.82, 129.31, 129.11, 128.70, 124.28, 123.00, 88.01. ESI-MS *m/z*: 453.74 [M + H]^+^, C_22_H_16_INO_2_ [453.02] (Appendix A).

(*E*)*-N-*(4-methoxyphenyl)-4-(3-oxo-3-phenylprop-1-en-1-yl) benzamide (**B5**). Eluent petroleum ether/EtOAc (7:1). Yellow solid, 40% yield, mp: 189.1–191.2 °C. ^1^H NMR (400 MHz, DMSO-*d_6_*) δ 10.22 (s, 1H), 8.24–8.15 (m, 2H), 8.05 (dd, *J* = 7.9, 3.8 Hz, 5H), 7.81 (d, *J* = 15.6 Hz, 1H), 7.72–7.67 (m, 3H), 7.60 (dd, *J* = 8.3, 6.9 Hz, 2H), 6.97–6.92 (m, 2H), 3.76 (s, 3H). ^13^C NMR (101 MHz, DMSO-*d_6_*) δ 189.63, 164.84, 156.11, 143.29, 137.88, 136.82, 133.80, 132.55, 129.32, 129.27, 129.10, 128.55, 124.10, 122.52, 114.94, 114.23, 55.65. ESI-MS *m/z*: 357.87 [M + H]^+^, C_23_H_19_NO_3_ [357.14] (Appendix A).

(*E*)-4-(3-oxo-3-phenylprop-1-en-1-yl)*-N-*(p-tolyl) benzamide (**B6**). Eluent petroleum ether/EtOAc (7:1). Yellow solid, 49% yield, mp: 192.3–194.2 °C. ^1^H NMR (400 MHz, DMSO-*d_6_*) δ 10.26 (s, 1H), 8.23–8.16 (m, 2H), 8.05 (dd, *J* = 8.7, 4.9 Hz, 4H), 7.81 (d, *J* = 15.6 Hz, 1H), 7.68 (t, *J* = 8.1 Hz, 3H), 7.61 (d, *J* = 7.5 Hz, 2H), 7.52 (dd, *J* = 17.5, 8.1 Hz, 1H), 7.17 (d, *J* = 8.2 Hz, 2H), 2.29 (s, 3H). ^13^C NMR (101 MHz, DMSO-*d_6_*) δ 192.81, 169.54, 145.52, 140.24, 139.53, 137.56, 135.70, 133.52, 132.20, 131.70, 131.68, 131.67, 131.42, 131.08, 126.34, 119.62, 21.89. ESI-MS *m/z*: 341.91 [M + H]^+^, C_23_H_19_NO_2_ [341.14] (Appendix A).

(*E*)-4-(3-oxo-3-phenylprop-1-en-1-yl)*-N-*(thiazol-2-yl) benzamide (**B7**). dichloromethane/methanol(100:1). White solid, 60% yield, mp: 229.3–231.8 °C. ^1^H NMR (400 MHz, DMSO-*d_6_*) δ 8.20 (t, *J* = 7.8 Hz, 4H), 8.08 (d, *J* = 8.7 Hz, 3H), 7.82 (d, *J* = 15.7 Hz, 1H), 7.70 (t, *J* = 7.3 Hz, 1H), 7.63–7.57 (m, 3H), 7.31 (d, *J* = 3.5 Hz, 1H). ^13^C NMR (101 MHz, DMSO-*d_6_*) δ 189.58, 142.98, 138.93, 137.83, 133.83, 129.39, 129.32, 129.13, 124.64, 114.44. ESI-MS *m/z*: 334.87 [M + H]^+^, C19H14N2O2S [334.08] (Appendix A).

(*E*)-4-(3-oxo-3-phenylprop-1-en-1-yl)*-N-*(thiophen-2-ylmethyl) benzamide (**B8**). dichloromethane/methanol(100:1). Yellow solid, 61% yield, mp: 145.9–147.1 °C. ^1^H NMR (400 MHz, DMSO-*d_6_*) δ 9.26 (t, *J* = 5.8 Hz, 1H), 8.18 (d, *J* = 7.3 Hz, 2H), 8.02 (d, *J* = 4.8 Hz, 1H), 8.00 (s, 1H), 7.96 (d, *J* = 8.4 Hz, 2H), 7.76 (t, *J* = 4.0 Hz, 1H), 7.72 (dd, *J* = 5.9, 3.2 Hz, 1H), 7.69 (d, *J* = 6.2 Hz, 1H), 7.59 (t, *J* = 7.6 Hz, 2H), 7.40 (dd, *J* = 5.1, 1.1 Hz, 1H), 7.04 (d, *J* = 2.4 Hz, 1H), 6.97 (dd, *J* = 5.0, 3.5 Hz, 1H), 4.66 (d, *J* = 5.8 Hz, 2H). ^13^C NMR (101 MHz, DMSO-*d_6_*) δ 192.81, 169.53, 145.52, 140.08, 139.53, 137.56, 134.83, 133.52, 131.97, 131.68, 131.42, 131.15, 129.71, 128.02, 127.59, 126.34, 46.98. ESI-MS *m/z*: 347.89 [M + H]^+^, C_21_H_17_NO_2_S [347.10] (Appendix A).

(*E*)-4-(3-oxo-3-phenylprop-1-en-1-yl)*-N-*((tetrahydrofuran-3-yl) methyl) benzamide (**B9**). dichloromethane/methanol(110:1). Yellow solid, 69% yield, mp: 103.6–105.2 °C. ^1^H NMR (400 MHz, DMSO-*d_6_*) δ 8.69 (t, *J* = 5.6 Hz, 1H), 8.18 (d, *J* = 7.2 Hz, 2H), 8.03–7.98 (m, 3H), 7.93 (d, *J* = 8.3 Hz, 2H), 7.79 (d, *J* = 15.7 Hz, 1H), 7.73–7.67 (m, 1H), 7.59 (t, *J* = 7.6 Hz, 2H), 3.72 (ddd, *J* = 20.8, 8.0, 6.5 Hz, 2H), 3.63 (q, *J* = 7.8 Hz, 1H), 3.50 (dd, *J* = 8.5, 5.3 Hz, 1H), 3.33–3.22 (m, 2H), 2.02–1.89 (m, 1H), 1.62 (dq, *J* = 12.6, 6.7 Hz, 1H). ^13^C NMR (101 MHz, DMSO-*d_6_*) δ 189.62, 166.21, 143.34, 137.89, 137.64, 136.37, 133.75, 129.29, 129.21, 129.07, 128.18, 123.93, 70.98, 67.27, 42.47, 29.98. ESI-MS *m/z*: 335.93 [M + H]^+^, C_21_H_21_NO_3_ [335.15] (Appendix A).

(*E*)*-N-*(cyclopropylmethyl)-4-(3-oxo-3-phenylprop-1-en-1-yl) benzamide (**B10**). dichloromethane/methanol(100:1). White solid, 58% yield, mp: 120.4–124.3 °C. ^1^H NMR (400 MHz, DMSO-*d_6_*) δ 8.44 (t, *J* = 5.5 Hz, 1H), 7.94 (d, *J* = 7.3 Hz, 2H), 7.83–7.69 (m, 5H), 7.56 (s, 1H), 7.45 (t, *J* = 7.3 Hz, 1H), 7.35 (t, *J* = 7.6 Hz, 2H), 2.93 (t, *J* = 6.2 Hz, 2H), 0.81 (tq, *J* = 12.9, 7.1, 6.0 Hz, 1H), 0.26–0.17 (m, 2H), 0.01 (q, *J* = 4.8 Hz, 2H). ^13^C NMR (101 MHz, DMSO-*d_6_*) δ 189.60, 165.89, 143.37, 137.90, 137.56, 136.47, 133.74, 129.28, 129.19, 129.07, 128.19, 123.89, 44.09, 11.47, 3.83. ESI-MS *m/z*: 328.88 [M + Na]^+^, C_20_H_19_NO_2_ [305.14] (Appendix A).

(*E*)-4-(3-oxo-3-phenylprop-1-en-1-yl)*-N-*(1H-pyrazol-5-yl) benzamide (**B11**). dichloromethane/methanol(120:1). Brown solid, 39% yield, mp: 85.6–87.3 °C. ^1^H NMR (400 MHz, DMSO-*d_6_*) δ 8.21 (dd, *J* = 10.2, 5.1 Hz, 3H), 8.05 (q, *J* = 8.3 Hz, 5H), 7.81 (d, *J* = 15.7 Hz, 1H), 7.70 (t, *J* = 7.3 Hz, 1H), 7.60 (t, *J* = 7.6 Hz, 2H), 6.06 (d, *J* = 2.9 Hz, 1H). ^13^C NMR (101 MHz, DMSO-*d_6_*) δ 189.66, 164.39, 159.98, 143.17, 138.33, 137.86, 134.50, 133.82, 131.83, 131.43, 129.33, 129.11, 128.66, 124.56, 102.91. ESI-MS *m/z*: 339.87 [M + Na]^+^, C_19_H_15_N_3_O_2_ [317.12] (Appendix A).

2-((1H-benzo[d]imidazol-2-yl)thio)ethyl (*E*)-4-(3-oxo-3-phenylprop-1-en-1-yl) benzoate (**B12**). petroleum ether/EtOAc (4:1). Yellow solid, 60% yield, mp: 152.0–154.2 °C. ^1^H NMR (400 MHz, Chloroform-d) δ 8.06 (d, *J* = 8.0 Hz, 2H), 8.00 (d, *J* = 8.2 Hz, 2H), 7.80 (d, *J* = 15.7 Hz, 1H), 7.64 (s, 3H), 7.59–7.53 (m, 4H), 7.28 (s, 1H), 7.25 (dd, *J* = 6.0, 3.2 Hz, 2H), 4.71 (t, *J* = 6.3 Hz, 2H), 3.71 (t, *J* = 6.3 Hz, 2H). ^13^C NMR (101 MHz, Chloroform-d) δ 192.81, 170.30, 155.44, 145.52, 142.19, 140.51, 139.53, 138.02, 133.52, 132.65, 131.68, 131.42, 131.20, 130.91, 127.78, 127.54, 126.34, 119.34, 116.12, 65.59, 33.85. ESI-MS *m/z*: 450.77 [M + Na]^+^, C_25_H_20_N_2_O_3_S [428.12] (Appendix A).

3-((1H-benzo[d]imidazol-2-yl) thio) propyl (*E*)-4-(3-oxo-3-phenylprop-1-en-1-yl) benzoate (**B13**). petroleum ether/EtOAc (3:1). Yellow solid, 59% yield, mp: 90.1–92.3 °C. ^1^H NMR (400 MHz, DMSO-*d_6_*) δ 8.37 (d, *J* = 5.8 Hz, 2H), 8.18 (d, *J* = 7.4 Hz, 2H), 8.05 (s, 5H), 7.79 (d, *J* = 15.7 Hz, 1H), 7.70 (t, *J* = 7.3 Hz, 1H), 7.60 (t, *J* = 7.6 Hz, 2H), 7.31 (d, *J* = 6.1 Hz, 2H), 4.41 (t, *J* = 6.1 Hz, 2H), 3.25 (t, *J* = 7.2 Hz, 2H), 2.11 (p, *J* = 6.6 Hz, 2H). ^13^C NMR (101 MHz, DMSO-*d_6_*) δ 192.81, 169.53, 155.73, 145.52, 141.49, 141.25, 139.53, 138.02, 133.52, 133.32, 131.68, 131.42, 131.20, 130.79, 127.78, 127.54, 126.34, 119.34, 116.12, 65.94, 32.50, 28.97. ESI-MS *m/z*: 442.98 [M + H]^+^, C_26_H_22_N_2_O_3_S [442.14] (Appendix A).

3-(pyridin-4-ylthio) propyl(*E*)-4-(3-oxo-3-phenylprop-1-en-1-yl) benzoate (**B14**). petroleum ether/EtOAc (4:1). Yellow solid, 55% yield, mp: 105.7–107.8 °C. ^1^H NMR (400 MHz, Chloroform-d) δ 8.41 (d, *J* = 5.5 Hz, 2H), 8.10 (d, *J* = 8.3 Hz, 2H), 8.05 (d, *J* = 7.2 Hz, 2H), 7.83 (d, *J* = 15.7 Hz, 1H), 7.73 (d, *J* = 8.3 Hz, 1H), 7.66–7.59 (m, 3H), 7.53 (t, *J* = 7.6 Hz, 1H), 7.15 (d, *J* = 6.3 Hz, 2H), 4.50 (t, *J* = 6.1 Hz, 2H), 3.16 (t, *J* = 7.2 Hz, 2H), 2.22 (p, *J* = 6.6 Hz, 2H). ^13^C NMR (101 MHz, Chloroform-d) δ 190.12, 149.36, 143.05, 137.87, 133.11, 130.17, 129.51, 129.22, 128.99, 128.75, 128.57, 128.32, 120.81. ESI-MS *m/z*: 403.97 [M + H]^+^, C_24_H_21_NO_3_S [403.12] (Appendix A).

### 3.3. Biological Evaluation

#### 3.3.1. CCK-8 Method for Evaluate Antiproliferative Activity

Cells were seeded into 96 well plates with 4000 cells/well using DMEM solution and incubated in a humid atmosphere containing 5% CO_2_ at 37 °C for 24 h. Compounds with different concentrations (200 μM, 100 μM, 50 μM, 25 μM, 12.5 μM, 6.25 μM, 3.125 μM, 1.5625 μM) were added to the culture medium for 24 h. After that, CCK-8 reagent (10 μL) was added under dark conditions and incubated the plate in a CO_2_ incubator at 37 °C for 2 h. After incubation, the absorbance was measured using an enzyme-linked immunosorbent assay at a wavelength of 450 nm. Cell survival rate was calculated as a percentage of the control group.

By using Graph Pad Prism 8.0 software, the concentration that inhibits 50% of maximum cell proliferation was determined to be IC_50_ value.

#### 3.3.2. Migration Ability Assay

MCF-7 cells were seeded into six-well plates at a density of 3000 cells/well and growing for 24 h. Scratches were made in confluent monolayers with 200 μL pipette tip. After that, each well was washed with PBS buffer 3 times to remove non-adherent cell debris. MCF-7 cells were continually incubated with different concentrations of compounds or positive control for 24 or 48 h. The area of cells migrated across the wound was photographed using phase contrast microscopy at 0 h, 24 h, and 48 h. The migration distance of cells into the wound area was measured by ImageJ2 software at room temperature.

#### 3.3.3. Cell Colony Formatting Assay

MCF-7 cells were seeded into six-well plates at a density of 1000 cells/well and cultivated for 24 h. After that, the cells were continually incubated with different concentrations of compounds and positive control for 7 days. Each well was carefully rinsed three times with PBS buffer quickly at room temperature. Collected cells were fixed for 1 h at room temperature by adding 2 mL of 4% paraformaldehyde solution. The results were analyzed using the ImageJ software. 

#### 3.3.4. Molecular Docking Study

The target protein was downloaded from the PDB database (https://www.rcsb.org/) (accessed on 18 March 2023), and the crystal structure of the target protein was processed by AutoDock Vina software (http://vina.scripps.edu/) (accessed on 18 March 2023). The water and modified amino acids were removed from the protein. The energy was optimized, and force field parameters were adjusted to meet a low energy conformation of the ligand structure. Three-dimensional structures of compounds were constructed using Chem3D software (14.0.0.17)and converted into pdbqt format for use. The target structure was then molecularly docked to the target compound using vina within pyrx software (0.8), and the affinity (kcal/mol) value was calculated. The lower the affinity value, the more stable the ligand-receptor binding. Finally, the results were visualized and analyzed using Pymol (1.8.x).

## 4. Conclusions

In this study, a series of novel chalcone derivatives were synthesized, and their antitumor activity was investigated. The compounds’ antiproliferative activity against breast cancer was evaluated through a CCK-8 assay. These compounds with trimethoxy phenyl structure in A series exhibited better in vitro cytotoxicity than 5-Fu. In addition, the introduction of the heterocyclic ring or oxygen alkylation attached in the A and B series resulted in enhanced cytotoxicity against MCF-7 cells. Compounds with good activity were selected for normal hepatocyte toxicity assay, scratch migration assay, and cell clone formation assay to evaluate their potential as antitumor candidates. The results showed that compound **A14** demonstrated significant inhibition of the proliferation and migration of breast cancer cells, and molecular docking results indicated that **A14** was able to bind with colchicine binding sites of microtubulin. It may serve as a new template for the synthesis and development of anti-breast cancer chemotherapy.

## Data Availability

Data is contained within the article and Appendix A.

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
