# Peer review of "Design and Synthesis of Novel Chalcone Derivatives: Anti-Breast Cancer Activity Evaluation and Docking Study"

_ijms, 2023, doi:10.3390/ijms242115549_

Round 1
Reviewer 1 Report
This paper describes synthesis of chalcone derivatives where the modifications were done in both A and B rings. Authors applied well known method of chalcone system construction applying Claisen-Schmidt condensation reaction (Eur. J Med Chem 37 (2002) 961-972; Eur. J Med Chem 66, (2013) 22-31). The appropriate donor, namely 3,4,5-trimethoxyacetophenone or acetophenone was condensed with 4-formylbenzoic acid. The product, α, β-unsaturated ketone containing in ring B carboxylic group was further transformed into corresponding acyl chloride and treated with appropriate primary amines. The appropriate amides were obtained in moderate yield. Another set of derivatives was obtained in reaction of chalcone carboxylic acid with dibromo alkanes, and treatment the formed ester with different thiols. Among twenty-eight synthesized derivatives four of them exhibit IC50 below 20 µMol. The preparation methods applied for the synthesis of both series A and B is well known and did not exhibit any novelty. The biological investigation contains standard assessments of cytotoxicity and clonogenic potential enriched by inhibition of migration ability and docking study between compound A14 and the colchicine binding site of the tubulin crystal structure. This docking study is low informative; the distance is reported between A14 and arginine moieties as a full molecule. Questions is which atoms of arginine participate in these interactions? The binding free energy of compound A14 were -7.2 kcal/mol, indicating moderate affinity towards tubulin. It will be helpful to compare this data with combretastatin A-4 or colchicine a known tubulin inhibitor.
Proper name for reactant 3 is 4-formylbenzoic acid (page 8 line 196). For compounds 2a, 4a, 4b, 5b, 5c, 5e, 5f 13C NMR spectra are not present. The manufacturer of CCK-8 kit should have to be indicated. For all solids, the melting point must be reported. In the case of known derivatives, the reported mp. must be added. The information about the sources of amine and mercaptans derivatives are lost (substituents present in scheme 2).
In general manuscript requires proof-reading by English native speaker.
In several phrases English is difficult for understanding e.g.,
Page 1 line 35: Most chalcone derivatives were designed mainly based on substitution of aryl rings, conjugation with other pharmacologically heterocyclics or residue modification of two aryl groups.
Page 2 line 46: Thus, microtubulin has considered an important target for cancer chemotherapy as for they can arrest tumor cell’s continuous abnormal proliferation.
Page 7 line 158 Previous studies reported that 3,4,5-trimethoxyphenyl unit as a potent fragment to interact tubulin.
Author Response
Response to Reviewer 1 Comments
Dear reviewer:
We would like to extend our most sincere appreciation to your valuable comments. According to the comments, we have improved the quality of our work, and highlighted the changes with red text in the manuscript, please see below for a point -by-point response to the comments.
Comment 1: This paper describes synthesis of chalcone derivatives where the modifications were done in both A and B rings. Authors applied well known method of chalcone system construction applying Claisen-Schmidt condensation reaction (Eur. J Med Chem 37 (2002) 961-972; Eur. J Med Chem 66, (2013) 22-31). The appropriate donor, namely 3,4,5-trimethoxyacetophenone or acetophenone was condensed with 4-formylbenzoic acid. The product, α, β-unsaturated ketone containing in ring B carboxylic group was further transformed into corresponding acyl chloride and treated with appropriate primary amines. The appropriate amides were obtained in moderate yield. Another set of derivatives was obtained in reaction of chalcone carboxylic acid with dibromo alkanes, and treatment the formed ester with different thiols. Among twenty-eight synthesized derivatives four of them exhibit IC50 below 20 µMol. The preparation methods applied for the synthesis of both series A and B is well known and did not exhibit any novelty.
Response 1: thank you for your professional comment. Although our synthesis methods were not novel,but our design and synthesised 28 novel chalcone derivitives never reported before. And compound A14 demonstrated significant inhibition of proliferation and migration of breast cancer cells. It maybe serve as a new template for the development of anti-breast cancer chemotherapy
Comment 2: The biological investigation contains standard assessments of cytotoxicity and clonogenic potential enriched by inhibition of migration ability and docking study between compound A14 and the colchicine binding site of the tubulin crystal structure. This docking study is low informative; the distance is reported between A14 and arginine moieties as a full molecule. Questions is which atoms of arginine participate in these interactions? The binding free energy of compound A14 were -7.2 kcal/mol, indicating moderate affinity towards tubulin. It will be helpful to compare this data with combretastatin A-4 or colchicine a known tubulin inhibitor.
Response 2: thank you for your question. According to the docking mode, The oxygen atom in 3,4,5-trimethoxyphenyl of compound A14 forms a hydrogen bonding interaction with hydrogen in secondary amine of Arg-61(2.3 Å), the carbonyl of α,β-unsaturated carbonyl system establishes hydrogen bonds with hydrogen atom in primary amine of Arg 164(2.7 Å ) and Arg 105(2.1 Å), and we added the binding data of combretastatin A-4.(section 2.3)
Comment 3: Proper name for reactant 3 is 4-formylbenzoic acid (page 8 line 196).
Response 3: We are really sorry for our careless mistakes. Thank you for your reminder. We have corrected “para-aldehyde benzoic acid ” to “4-formylbenzoic acid”.(page14 line 99)
Comment 4:For compounds 2a, 4a, 4b, 5b, 5c, 5e, 5f 13C NMR spectra are not present.
Response 4: We are so sorry for our negligence. And we have added 13C NMR spectra in supporting information and added the analysis data in manuscript.
Comment 5: The manufacturer of CCK-8 kit should have to be indicated.
Response 5: Thank you for your careful suggestion. According to your comment, we have added the manufacturer of CCK-8 kit, paraformaldehyde and Crystal Violet.
Comment 6: For all solids, the melting point must be reported. In the case of known derivatives, the reported mp. must be added.
Response 6: We sincerely grateful for your valuable comment, and we have added the melting point of all solids.
Comment 7: The information about the sources of amine and mercaptans derivatives are lost (substituents present in scheme 2).
Response 7: Thank you for your careful checks. We actually synthesised 28 compounds. All the primary amine compounds and thiol compounds were purchased from Macklin.
Comment 8: In general manuscript requires proof-reading by English native speaker.
Response 8: thank you for your suggestion. We feel sorry for our poor writings, we tried our best to improve the manuscript and made some changes to the manuscript. And we did not list the changes but marked in the revised paper. We sincerely appreciate for reviewers’ warm work earnestly anf hope that the correction will meet with approval.
Comments on the Quality of English Language
In several phrases English is difficult for understanding e.g.,
Comment 9: Page 1 line 35: Most chalcone derivatives were designed mainly based on substitution of aryl rings, conjugation with other pharmacologically heterocyclics or residue modification of two aryl groups.
Response 9: we are very sorry for the mistakes in this manuscript and incovenience it caused in your reading, this sentence have been corrected to “The design methods of most chalcone derivatives mainly include 1) substitution of aromatic rings; 2) Binding with other pharmacological heterocycles; 3) Residue modification of two aryl groups”(page 4 line 35)
Comment 10: Page 2 line 46: Thus, microtubulin has considered an important target for cancer chemotherapy as for they can arrest tumor cell’s continuous abnormal proliferation.
Response 10: we are very sorry for the mistakes in this manuscript and incovenience it caused in your reading, this sentence have been corrected to “Thus, microtubulin is considered an important target for cancer chemotherapy because of it’s ability of arrest tumor cell’s continuous abnormal proliferation ”(page 4 line 46)
Comment 11:Page 7 line 158 Previous studies reported that 3,4,5-trimethoxyphenyl unit as a potent fragment to interact tubulin.
Response 11: we are very sorry for the mistakes in this manuscript and incovenience it caused in your reading, this sentence have been corrected to”Previous studies indicated that 3,4,5-trimethoxyphenyl unit as a potent fragment to interact with tubulin ”.(page 12 line 149)

Reviewer 2 Report
Chen, Ye, et al. described preparation of novel chalcones and evaluation of their anti-breast cancer activity.
The work is interesting but needs much improvements before being considered for publication.
English language must be checked by native English researcher.
All new intermediates and new final compounds must be fully characterized : 13C NMR, HRMS or elemental analysis.
Standard deviations should be provided in table 1.
Check caption of figures 3 and 4.
First line of scheme 1 must be checked: no OH for 2a; + 3 beside 2a is confusing; reagents and conditions should appear scheme 1 and not scheme 2;
Bibliography should be formatted as for IJMS.
Minor remarks: check bold numbers for compounds; chalcone (first word of the abstract) not in bold
Line 96: define 5-FU at its first appearance.
Check captions in SI
English language must be checked by native English researcher.
Author Response
Response to Reviewer 2 Comments
Dear reviewer:
We would like to extend our most sincere appreciation to your valuable comments. According to the comments, we have improved the quality of our work, and highlighted the changes with red text in the manuscript, please see below for a point -by-point response to the comments.
Comment 1: Chen, Ye, et al. described preparation of novel chalcones and evaluation of their anti-breast cancer activity. The work is interesting but needs much improvements before being considered for publication. English language must be checked by native English researcher.
Response 1: Thank you for your suggestion. We feel so sorry for our poor writings, we tried our best to improve the manuscript and made some changes to the manuscript. And we did not list the changes but marked in the revised paper. We sincerely appreciate for reviewers’ warm work earnestly anf hope that the correction will meet with approval.
Comment 2: All new intermediates and new final compounds must be fully characterized : 13C NMR, HRMS or elemental analysis.
Response 2: We sincerely thank you for your suggestion. In our synthesis section, we confirmed the strutures of intermediates by 1H and we then added the 13 C NMR spectura, and then continue to use for the next synthesis until obtain final products, And the strutures of final products were both correct. Therefore, for the intermediates, we confirmed its structure through 1H and 13 C NMR.
Comment 3: Standard deviations should be provided in table 1.
Response 3: Thank you for your careful checks. We have calculated the Standard deviations and added in table 1.
Comment 4: Check caption of figures 3 and 4.
Response 4: We are grarefully appreciate for your professional comment. We have corrected the caption of figure4.(page 11 line 138)
Comment 5: First line of scheme 1 must be checked: no OH for 2a; + 3 beside 2a is confusing; reagents and conditions should appear scheme 1 and not scheme 2;
Response 5: Thank you for your careful checks. We have draw the correct struture of 2a, the synthesis method of 4a was compound 2a reacted with compound 3, and we have attached new scheme 1 in manuscript. We have piaced the description of the reaction conditions in correct site. (page 7 line 84)
Comment 6: Bibliography should be formatted as for IJMS.
Response 6: We thank reviewer for the valuable suggestion, and we have checked the reference and the format according to MDPI IJMS format.(page 27-33)
Comment 7: Minor remarks: check bold numbers for compounds; chalcone (first word of the abstract) not in bold
Response 7: Thank you for pointing this out. We are so sorry for our negligence of the content, We have throughly checked and corrected the content. We corrected “A15” to “A14”(page 12 line 160, page 11, line 138). we have bold the chalcone (first word of the abstract).
Comment 7: Line 96: define 5-FU at its first appearance.
Response 7: We are really sorry for our careless mistakes. And we have added the define of 5-Fu.(page 2 line 14)
Comment 8: Check captions in SI
Response 8: Thank you for pointing this out. We are so sorry for our negligence of the content, We have throughly checked and corrected the SI. All the changes were marked in the revised paper.
Comment 9: Comments on the Quality of English Language
English language must be checked by native English researcher.
Response 9: Thank you for your suggestion. We regret there were problems with the English. We have asked for professional for help.

Reviewer 3 Report
Review of "Design and Synthesis of Novel Chalcone Derivatives: Anti-breast cancer Activity Evaluation and Docking Study" by Ye et al.
The article concerns the synthesis of new chalcone derivatives with anti-cancer properties against MCF-7 breast cancer cells and docking studies were performed. The idea of the article is good, but it is not free from errors. Both stylistic, editorial and in the sphere of research.
Major remarks:
1) In the abstract lines 10-12: "In previous study, the research group discovered a series of chalcone derivatives with stronger anticancer activities." - please specify in which previous studies, citing e.g.
2) The citation [13] is for dihydrochalcones and not chalcones. Please correct or give a different citation.
3) Line 39: "meth chalcone" - should be "metochalcone". Especially, that further authors already spell "meochalcone" correctly
4) LINE 39-40: missing citation for properties of metochalcone and sofalcone. Please add.
5) Line 58: "3,4,5-trimethoxyphenylis" - should be "3,4,5-trimethoxyphenyl"
6) Lines 57-59: scaffold B and scaffold A - should be changed places in the text. I understand that in Fig. 1, scaffold A is the basic chalcone structure and Scaffold B is the 3,4,5-timethoxy substituted structure. But in Fig 2, I have the impression that A - is the basic structure and B is the 3,4,5-trimethoxy basic structure. Please explain and possibly correct.
7) Line 72: "scheme 1 and scheme 2" should be "Scheme 1 and Scheme 2."
8) Line 73: "Foucault-acylation" should be "Friedel-Crafts reaction or acylation".
9) Line 80: "O-alkoxylate" should be "O-alkoxylate" (italic O). Throughout the article, please correct such things, both in the abbreviation of the para substitutions "p" or "N,N-" on lines 207 and 213, 232, etc.
10) Line 84-85: FT-IR spectra are missing for full characterization of compounds, please complete - add absorption bands to section 3.2 and spectra to supplementary materials.
11) Lines 90-92: the description of the reaction conditions should be placed directly under line 87 (under the name of Scheme 1.)
12) Table 2. Why did not the authors calculate the Selectivity Index (SI) for these compounds and did not discuss them? Additionally, it could be compared with the SI for 5-Fu. I think it's worth adding.
13) Table 1. In my opinion compounds B8, B9 B10 as analogs of compounds A7, A8, A9, respectively, also works well. Why for B8, B9 B10 there were no tests with L-O2. Maybe despite the higher IC50 against the MCF-7 line for these compounds, SI would turn out to be better? Please explain and add any additional information.
14) Line 158: "previous studies" - missing citation. Please add.
15) Section 3.1. - please add specific device names (models) and the producers of the materials and equipment used, as well as the city and state. Please also provide the ATCC or other number of the cell lines used.
16) Section 3.2 Please provide specific elements that were used to purify specific compounds (can be provided with spectroscopic characteristics, preferably with Rf coefficients)
17) Synthesis on line 187: In what atmosphere was the synthesis carried out? Please add.
18) Line 200: "diluted hydrochloric acid" - please give specific percentage or molar concentration.
19) If an anhydrous solvent was used in any synthesis - please add it in the recipe.
20) "(E)" in the names of chalcones should be written in italics.
21) Many compounds (2a, 4a, 4b, 5b, 5c, 5e, 5f) lack 1H and 13H NMR spectra as MS. They also lack melting points. Even if they are already known. Please complete the.
22) Line 476-477: please correct these comma marks.
23) Line 475-476: "Compounds with different concentrations" - in what solvent were the compounds dissolved. Please add.
24) Sections 3.3.2 and 3.3.3 - at what temperature were the tests carried out? Please specify.
25) What was the purity of the compounds used in the biological research? Please add.
26) Please correct the Reference section according to MDPI IJMS format. Also, please correct the format of citations in the text.
The English language requires a lot of revision, preferably by a native speaker who specializes in scientific texts.
Author Response
Response to Reviewer 3 Comments
Dear reviewer:
We would like to extend our most sincere appreciation to your valuable comments. According to the comments, we have improved the quality of our work, and highlighted the changes with red text in the manuscript, please see below for a point -by-point response to the comments.
Comments 1: In the abstract lines 10-12: "In previous study, the research group discovered a series of chalcone derivatives with stronger anticancer activities." - please specify in which previous studies, citing e.g.
Response 1: Thank you for pointing this out. We are so sorry for our negligence of the explanation, and we have citted corresponding reference.(page 2, line 12)
Comments 2:The citation [13] is for dihydrochalcones and not chalcones. Please correct or give a different citation.
Response 2: Thank for your comments. We have throughly checked and corrected the reference and content.
Comments 3:Line 39: "meth chalcone" - should be "metochalcone". Especially, that further authors already spell "meochalcone" correctly
Response 3: We are really sorry for our careless mistakes. Thank you for your reminder. We have corrected “meth chalcone” to “metochalcone”.((page 4, line 39)
Comments 4: LINE 39-40: missing citation for properties of metochalcone and sofalcone. Please add.
Response 4: We are so sorry for our negligence of this sentence. We have added correct reference.
Comments 5: Line 58: "3,4,5-trimethoxyphenylis" - should be "3,4,5-trimethoxyphenyl"
Response 5: We are really sorry for our careless mistakes. Thank you for your reminder. We have corrected “3,4,5-trimethoxyphenylis” to “3,4,5-trimethoxyphenyl”.((page 5, line 58)
Comment 6: Lines 57-59: scaffold B and scaffold A - should be changed places in the text. I understand that in Fig. 1, scaffold A is the basic chalcone structure and Scaffold B is the 3,4,5-timethoxy substituted structure. But in Fig 2, I have the impression that A - is the basic structure and B is the 3,4,5-trimethoxy basic structure. Please explain and possibly correct.
Response 6: Thank for your carefully comment. For our research, scaffold A is the 3,4,5-timethoxy substituted structureand Scaffold B is the basic chalcone structure. and we have reedited figure 2.(page 5)
Comment 7: Line 72: "scheme 1 and scheme 2" should be "Scheme 1 and Scheme 2."
Response 7: Thank you for pointing this out. We have corrected “scheme 1 and scheme 2" to "Scheme 1 and Scheme 2”according to your reminder.(page 6 line 70)
Comment 8: Line 73: "Foucault-acylation" should be "Friedel-Crafts reaction or acylation".
Response 8: We are really sorry for our careless mistake. Thank you for your reminder. We have corrected “Foucault-acylation” to “Friedel-Crafts reaction”.((page 6, line 71)
Comment 9: Line 80: "O-alkoxylate" should be "O-alkoxylate" (italic O). Throughout the article, please correct such things, both in the abbreviation of the para substitutions "p" or "N,N-" on lines 207 and 213, 232, etc.
Response 9: We are really sorry for our careless mistakes. Thank you for your reminder. We have throughly checked and corrected the formats. We have corrected “O-alkoxylate” to “O-alkoxylate”(page 6 line 76), and also corrected the “para-aldehyde benzoic acid” to “4-formylbenzoic acid”(page 14 line 195); The font Style of “N,N-” were also corrected to “N,N-”.(page 14 line 206, page 15 line 212 and line 232)
Comment 10: Line 84-85: FT-IR spectra are missing for full characterization of compounds, please complete - add absorption bands to section 3.2 and spectra to supplementary materials.
Response10: We appreciate for your kind suggestion. We have confirmed all target compounds by 1H NMR, 13C NMR and mass spectrum, so we didn’t added FT-IR spectra. Some research also confirmed compound’s sturtures by 1H NMR, 13C NMR and mass spectrum but without FT-IR spectra[1,2].
Comment 11: Lines 90-92: the description of the reaction conditions should be placed directly under line 87 (under the name of Scheme 1.)
Response 11: Thank for your carefully comment. We have piaced the description of the reaction conditions in page 7 (under the name of Scheme 1).(page 7)
Comment 12: Table 2. Why did not the authors calculate the Selectivity Index (SI) for these compounds and did not discuss them? Additionally, it could be compared with the SI for 5-Fu. I think it's worth adding.
Response 12: We are grarefully appreciate for your valuable comment. We have added the Selectivity Index (SI) for these four compounds and discuss them, we also compared with the SI for 5-Fu. (page 10 table 2)
Comment 13: Table 1. In my opinion compounds B8, B9 B10 as analogs of compounds A7, A8, A9, respectively, also works well. Why for B8, B9 B10 there were no tests with L-O2. Maybe despite the higher IC50 against the MCF-7 line for these compounds, SI would turn out to be better? Please explain and add any additional information.
Response 13: We appreciate it very much for this good suggestion, When we tested compounds, we mainly consider the IC50 of series B compounds is relatively higher, so we didm’t discuss them more, and the later experiments also confirm that A14 was the most meaningful compound for further research. We have collect your sugggestion for further study.
Comment 14: Line 158: "previous studies" - missing citation. Please add.
Response 14: We are so sorry for our negligence of this sentence. We have added correct reference. (page 12 line 150)
Comment 15: Section 3.1. please add specific device names (models) and the producers of the materials and equipment used, as well as the city and state. Please also provide the ATCC or other number of the cell lines used.
Response 15: We sincerely grateful for your valuable comment.we have added the device names and the producers of the materials and equipment used.(page 13)
Comment 16: Section 3.2 Please provide specific elements that were used to purify specific compounds (can be provided with spectroscopic characteristics, preferably with Rf coefficients)
Response 16: Thank you for this valuable suggestion. We have been added corrseponding elution solvent and its ratio which used for purified compounds via silica gel column chromatography.(section 3.2)
Comment 17: Synthesis on line 187: In what atmosphere was the synthesis carried out? Please add.
Response 17: Thank you for your careful checks. On this point, This reaction was carried out under reflux. We have made the corrections.(page 14 line 200)
Comment 18: Line 200: "diluted hydrochloric acid" - please give specific percentage or molar concentration.
Response 18: we were really sorry for our careless mistakes. Thank you for your reminder. We have added the molar concentration of diluted hydrochloric acid in manuscript.(page 14 line 202)
Comment 19: If an anhydrous solvent was used in any synthesis - please add it in the recipe.
Response 19: Thank you for your careful checks, we have added anhydrous solvent which used in synthesis.(page 14 line 191 and 208;page 15 line 224 and 235)
Comment 20:"(E)" in the names of chalcones should be written in italics.
Response 20: Thank you for your careful checks, we have made the corrections on manuscript.(section 3.2)
Comment 21: Many compounds (2a, 4a, 4b, 5b, 5c, 5e, 5f) lack 1H and 13H NMR spectra as MS. They also lack melting points. Even if they are already known. Please complete the.
Response 21: Thank you for your suggestion. In our synthesis section, we confirmed the strutures of intermediates by 1H and we added the 13 C NMR spetura, and then continue to use for the next synthesis until obtain final products, which further proof that the intermediates structures were no question.
Comment 22: Line 476-477: please correct these comma marks.
Response 22: Thank you for your careful checks, we have made the corrections on manuscript.
Comment 23: Line 475-476: "Compounds with different concentrations" - in what solvent were the compounds dissolved. Please add.
Response 23: Thank you for your careful suggestion. According to your comment, we have added the solvent and cell culture conditions in section3.1.2.
Comment 24: Sections 3.3.2 and 3.3.3 - at what temperature were the tests carried out? Please specify.
Response 24: Thank you for your careful suggestion. All cells were cultivated at 37℃ under 5 % CO2. Observation of results were in room temperature.(page 25 line 507 and 512)
Commet 25 :What was the purity of the compounds used in the biological research? Please add.
Response 25: we sincerely thank you for the professional commet. We have added the purity of compound A14.
Commet 26: Please correct the Reference section according to MDPI IJMS format. Also, please correct the format of citations in the text.
Response 26: we thank reviewer for the valuable suggestion, and we have checked the reference and the format sccording to MDPI IJMS format.(page 27-33)
- Li, G.; Li, J.; Wang, W.; Feng, X.; Yu, X.; Yuan, S.; Zhang, W.; Chen, J.; Hu, C. Synthesis, In Vitro, and In Vivo Investigations of Pterostilbene-Tethered Analogues as Anti-Breast Cancer Candidates. Int J Mol Sci 2023, 24, doi:10.3390/ijms241411468.
- Liu, F.; Cao, X.; Zhang, T.; Xing, L.; Sun, Z.; Zeng, W.; Xin, H.; Xue, W. Synthesis and Biological Activity of Myricetin Derivatives Containing Pyrazole Piperazine Amide. Int J Mol Sci 2023, 24, doi:10.3390/ijms241310442.

Round 2
Reviewer 1 Report
The referee report please find in an attached file.

Author Response
Dear reviewer:
We would like to extend our most sincere appreciation to your valuable comments. According to the comments, we have improved the quality of our work, and highlighted the changes with blue text in the manuscript, please see below for a point -by-point response to the comments.
Comment 1: First, the developing systems for TLC for particular reactions should be indicated.
Response 1: Thank you for your professional comment. We added the corrseponding elution solvent and its ratio which used for purified each compounds via silica gel column chromatography.(page 16 line 244, page 16 line 263.)
Comment 2: Page 3; in sentence “Microtubules are polymers which formed” please remove word “which”.
Response 2: We sincerely grateful for your valuable comment. We have removed the word”which”.(page 4 line 42.)
Comment 3: Page 3; exchange [29].[30] by [29, 30].
Response 3: Thank you for your careful checks. We have corrected [29].[30] to [29, 30].(page 4 line 48)
Comment 4: Page 3; remove additional dot point after [36].
Response 4: We were really sorry for our careless mistakes. We have removed additional dot point after [36]. (page 5 line 53.)
Comment 5: Page 4; phrase “which formed the crude intermediate 5a under oxalyl chloride conditions and catalysed by DMF” I suggest replacing by “which formed the crude intermediate 5a under oxalyl chloride treatment in the presence of catalytic amounts of DMF”.
Response 5: Thank you for your suggestion. We have corrected the phrase according your valuable suggestion.(page 6 line 75)
Comment 6: In Table 1 and 2 a standard deviation should be denoted as ± for example: 123.3±1.63.
Response 6: We are so sorry for our negligence of this sentence. We have corrected the right form according your comment.
Comment 7: Page 7; A7, A8, A9 and A14 in bold
Response 7: We are very sorry for the mistakes in this manuscript and incovenience it caused in your reading, we have made these words bold.(page 9 line 115.)
Comment 8: Page8; A8 in bold
Response 8: Thank you for your careful checks,we have corrected “A8”in bold.(page 9 line 123)
Comment 9: Page 9; phrase “indicating compound A14 can also binding affinity towards tubulin”. Please replace by “indicating binding affinity of compound A14 towards tubulin”.
Response 9: We sincerely grateful for your valuable comment. We have made the corrections on manuscript.(page 11 line 154.)
Comment 10: Page 10; 3.1.1. Chemistry in capital letters
Response 10: We are really sorry for our careless mistakes. we have made the corrections on manuscript.(page 12 line 167)
Comment 11: Page 10; phrase “and gel 60–120 mesh was used for column chromatography silica”. Please exchange by “and silica gel 60–120 mesh was used for column chromatography.”
Response 11: Thank you for your careful checks, we have made the corrections on manuscript.(page 12 line 170)
Comment 12: Page 10; Phrase “The organic solvent was concentrated under reduced pressure”. Please exchange by “The organic phase was concentrated under reduced pressure.”
Response 12: Thank you for your careful checks, we have made the corrections on manuscript.( page 13 line 194)
Comment 13: In description of synthetic procedures should be indicated anhydrous solvent when it is necessary.
Synthesis of (2a), (5a), (A1) anhydrous DCM should be use.
Response 13: We are really sorry for our careless mistakes. Thank you for your reminder. we have made the corrections on manuscript.(page 13 line 190,207. page 14 line 227,)
Comment 14: Synthesis of (4a) phrase “at 50℃ for 5 hours under reflux” please replace by “at 50℃ for 5 hours under reflux condenser” Methanol has boiling point over 50 OC.
Response 14: Thank you for your careful suggestion. we have made the corrections on manuscript.(page 13 line 200)
Comment 15: In preparation of (5b) the base amount is not indicated.
Response 15: Thank you for your careful suggestion, we have added the density of dibromoethane. (page 14 line 217)
Comment 16: Page 11 preparation of (A1); phrase “under dry conditions” please replace by “under anhydrous conditions”.
Response 16: Thank you for your careful suggestion. we have made the corrections on manuscript.(page 14 line 227)
Comment 18: Word “eluent” is lost in protocols for A6 – A14 and B7 – B15.
Response 18: Thank you for your careful suggestion. We have added the elution solvent and its ratio .(page 14-22.)

Reviewer 2 Report
Despite the efforts of the authors, the work still needs many improvements before considering for publication.
How were the standard deviations calculated? Why only “+” and not “±”?
HRMS are not given. 4-5 decimals digits are required.
Intermediates remain not fully characterized (HRMS missing)
Synthesis 3.2 (synthesis of compounds) is a mess, names of compounds do not correspond to the numbering. See for example compound 2a (page 12).
13C NMR spectra do not correspond to the name of the compound. See for example 4a: false name, good spectrum.
There are many other examples.
Scheme 2: R1 and R2 do not correspond to scheme 1.
For R1, there is no need to draw CONH; for R2, its starts after the sulfur atom.
1H NMR spectra indicate that some compounds are not pure at all. For example, extra aromatic peaks for A4. Why so many peaks for methyl groups in A5? Many extra peaks for A6? Why a peak at 9.3 for A7 (NH exchangeable?) ? Why two peaks around 8.6 for A8? …
Author Response
Dear reviewer:
We would like to extend our most sincere appreciation to your valuable comments. According to the comments, we have improved the quality of our work, and highlighted the changes with red text in the manuscript, please see below for a point -by-point response to the comments.
Comment 1: How were the standard deviations calculated? Why only “+” and not “±”?
Response 1: We are really sorry for our careless mistakes. Thank you for your reminder. Graph Pad Prism software was used for caculated the standard deviations, and only copied numbers when made table 1. We have made the corrections on table 1.(page 9 line 113.)
Comment 2: HRMS are not given. 4-5 decimals digits are required.
Response 2: Thank you for your careful suggestion. We are so sorry that due to the limited instrument, it need spend some time to supplement HRMS. And the mass spectrum of our experiments can also proof the structure of compounds.
Comment 3: Intermediates remain not fully characterized (HRMS missing)
Response 3: Thank you for your suggestion. We are so sorry the we did not supplyment the HRMS, but we added the MS spectra of intermediates. and in our synthesis section, we confirmed the strutures of intermediates and then continue to use for the next synthesis until obtain final products, which further proof that the intermediates structures were no question.
Comment 4: Synthesis 3.2 (synthesis of compounds) is a mess, names of compounds do not correspond to the numbering. See for example compound 2a (page 12).
Response 4: Thank you for your careful suggestion. We are really sorry for our careless mistakes. we have made the corrections on manuscript(page 13 line 189), and the names of compounds were named by chemdraw software.
Comment 5: 13C NMR spectra do not correspond to the name of the compound. See for example 4a: false name, good spectrum.
There are many other examples.
Response 5: thank you for your valuable suggestion. The compounds are named according to Chemdraw. And we have checked up the name of compounds again.
Comment 6: Scheme 2:R1 and R2 do not correspond to scheme 1.
For R1, there is no need to draw CONH; for R2, its starts after the sulfur atom.
Response 6: Thank you for your valuable suggestion. and we have reedited scheme 2.
Comment 7: 1H NMR spectra indicate that some compounds are not pure at all. For example, extra aromatic peaks for A4. Why so many peaks for methyl groups in A5? Many extra peaks for A6? Why a peak at 9.3 for A7 (NH exchangeable?) ? Why two peaks around 8.6 for A8?
Response 7: We sincerely grateful for your valuable comment. We acknowledgeed that some compounds do have impurities. But we have made efforts to improve purity of compounds, other spectra also can confirm the accuracy of compounds.

Reviewer 3 Report
I would like to thank the Authors for answers to questions and changes introduced.
Only editing things remain to be changed - sometimes there is a lowercase letter instead of a capital letter and the English language needs to be checked again.
The English language needs to be checked again.
Author Response
Dear reviewer:
We would like to extend our most sincere appreciation to your valuable comments. According to the comments, we have improved the quality of our work, and highlighted the changes with red text in the manuscript, please see below for a point -by-point response to the comments.
Comment 1: I would like to thank the Authors for answers to questions and changes introduced.
Only editing things remain to be changed - sometimes there is a lowercase letter instead of a capital letter and the English language needs to be checked again.
Response 1: we thank the reviewer for this valuable suggestion, and the whole manscript has been polished accordingly.
Comment 2: The English language needs to be checked again.
Response 2: Thank you for your valuable suggestion. And we have checked the manuscript again accordingly.

Round 3
Reviewer 2 Report
Despite the efforts of the authors, the manuscript may not be published in its present form.
Table 2 : standard deviations were not corrected (not +)
tables 1 and 2 : same number of digits for the values and the SDs
Names and numbering remain incorrect : eg compoud 4a is a trimethoxy derivative (line 270)
What is the meaning of l1-oxydaneyl (lines 280 and 286)?
Scheme 1 : for 2a, C=O and not C=OH;
for 4a, one methyl group is missing
Most importantly, biological assays have no value when performed on compounds that are not pure and not fully characterized. From a chemical point of view, full characterization (HRMS or CHN) are mandatory for any new compound.